# Targeting ITGβ3 to Overcome Trastuzumab Resistance through Epithelial–Mesenchymal Transition Regulation in HER2-Positive Breast Cancer

**DOI:** 10.3390/ijms25168640

**Published:** 2024-08-08

**Authors:** Asiye Busra Boz Er, Idris Er

**Affiliations:** 1Department of Medical Biology, Faculty of Medicine, Recep Tayyip Erdogan University, Rize 53200, Turkey; asiyebusra.bozer@erdogan.edu.tr; 2Department of Medical Biology, Faculty of Medicine, Karadeniz Technical University, Trabzon 61080, Turkey

**Keywords:** HER2-positive, ITGB3, Hedgehog, EMT

## Abstract

HER2-positive breast cancer, representing 15–20% of all breast cancer cases, often develops resistance to the HER2-targeted therapy trastuzumab. Unfortunately, effective treatments for advanced HER2-positive breast cancer remain scarce. This study aims to investigate the roles of ITGβ3, and Hedgehog signaling in trastuzumab resistance and explore the potential of combining trastuzumab with cilengitide as a therapeutic strategy. Quantitative gene expression analysis was performed to assess the transcription of EMT (epithelial–mesenchymal transition) markers *Slug*, *Snail*, *Twist2*, and *Zeb1* in trastuzumab-resistant HER2-positive breast cancer cells. The effects of ITGβ3 and Hedgehog signaling were investigated. Additionally, the combination therapy of trastuzumab and cilengitide was evaluated. Acquired trastuzumab resistance induced the transcription of *Slug*, *Snail*, *Twist2*, and *Zeb1*, indicating increased EMT. This increased EMT was mediated by ITGB3 and Hedgehog signaling. ITGβ3 regulated both the Hedgehog pathway and EMT, with the latter being independent of the Hedgehog pathway. The combination of trastuzumab and cilengitide showed a synergistic effect, reducing both EMT and Hedgehog pathway activity. Targeting ITGβ3 with cilengitide, combined with trastuzumab, effectively suppresses the Hedgehog pathway and EMT, offering a potential strategy to overcome trastuzumab resistance and improve outcomes for HER2-positive breast cancer patients.

## 1. Introduction

Breast cancer is the most common cause of cancer-related death in women worldwide [1], and approximately 20% of breast tumors exhibit HER2 overexpression. Trastuzumab, a HER2-targeting monoclonal antibody approved by the FDA (Food and Drug Administration), has shown promise in treating HER2-positive breast cancer [2]. However, the development of resistance to trastuzumab within a short timeframe leads to metastasis and a poor prognosis for patients [3,4]. An increasing body of research has focused on drug resistance to improve patient outcomes [5]. 

Acquired resistance to trastuzumab involves several mechanisms, including activation of integrin-mediated signaling. Integrins, which are cell surface receptors, facilitate interactions between cells and their environment, playing a crucial role in cancer growth and progression. Research has demonstrated their involvement in tumor development and drug resistance [6]. For instance, elevated levels of integrin α5β1 have been linked to doxorubicin resistance in MCF7 breast cancer cells [7]. Additionally, Seguin et al. identified that integrin αvβ3 is overexpressed in breast and lung cancer cells, where it collaborates with KRAS and RALB proteins to induce resistance to erlotinib [8,9]. These findings collectively highlight the significance of RGD-binding integrins in drug resistance across various cancer types. The mechanisms by which integrin mediates drug resistance are complex and multifaceted. Integrins play crucial roles in cell adhesion, migration, and resistance to therapy by the activation of key signaling pathways, such as MAPK [10], PI3K [11], and TGF-B [12]. These findings emphasize the importance of the role of RGD-binding integrins in trastuzumab resistance and mediating pathways to improve treatment strategies.

RGD-binding integrins also involve migratory changes such as EMT (epithelial–mesenchymal transition) [13,14]. EMT is a crucial process where epithelial cells lose their polarity and adhesion properties, gaining a mesenchymal phenotype that increases their migratory and invasive abilities. This transition is controlled by transcription factors like *Slug*, *Snail*, *Twist2*, and *Zeb1*, which are known to drive cancer metastasis and contribute to therapeutic resistance [15]. It was found that increased ITGβ3 enhances the migration and invasion abilities of breast cancer cells by promoting Snail expression and triggering EMT through the TGF-β pathway [12]. 

Our previous study showed that ITGβ3 regulates the Notch pathway in HER2-positive, trastuzumab-resistant, and parental breast cancer cell lines [16]. The Notch pathway is known for its crosstalk with the Hedgehog pathway [17], a key regulator of EMT and drug resistance [15,18]. Nevertheless, the mechanisms associated with Hedgehog signaling and ITGβ3 are poorly understood, as well as how they affect the EMT of cancer cells.

In summary, the aim of this study is to investigate the role of ITGβ3 and Hedgehog signaling in the development of trastuzumab resistance in HER2-positive breast cancer and to explore the potential of a combination therapy using trastuzumab and cilengitide to overcome this resistance. This study focuses on understanding how ITGβ3 regulates EMT and its impact on Hedgehog signaling and examines whether targeting ITGβ3 with cilengitide can suppress EMT and Hedgehog pathway activity, thereby potentially improving the treatment response and overcoming the resistance of patients with trastuzumab-resistant HER2-positive breast cancer.

## 2. Results

### 2.1. Chronic Trastuzumab Exposure Induces EMT in HER2-Positive Breast Cancer Cell Lines

To understand the initiation of metastasis associated with trastuzumab resistance in HER2-positive breast cancer, the expressions of EMT-activating transcription factors *Slug*, *Snail*, *Twist2*, and *Zeb1* [19] were analyzed in SKBR3 and HCC1954 cells over time. Both cell lines were treated daily with 0.05 μM trastuzumab for 2, 4, and 15 days, and gene expression was assessed on these days.

Significant decreases in *Slug*, *Snail*, *Twist2*, and *Zeb1* expressions were observed by day 2 in both SKBR3 and HCC1954 cells (Figure 1A,C). In SKBR3 cells, a continued decrease in all markers was noted by day 4, with no further changes by day 15. In HCC1954 cells, *Slug*, *Snail*, and *Twist2* decreased by day 4, while *Zeb1* remained unchanged; by day 15, no changes in any markers were observed.

To observe the long-term effects of trastuzumab exposure on EMT, trastuzumab-resistant HER2-positive SKBR3 and HCC1954 cell lines were used. In our previous work [16], these resistant cell lines were generated by exposing the cells to increasing doses of trastuzumab (0.1–10 μM) over 3 months, resulting in an approximate 8-fold increase in the IC_50_ value for HCC1954 and a 13-fold increase for SKBR3. In these resistant cells, significant increases in the expressions of *Slug*, *Snail*, *Twist2*, and *Zeb1* were observed (Figure 1B,D).

It was observed that chronic trastuzumab exposure initially decreases EMT marker expression in HER2-positive breast cancer cells. However, long-term exposure leads to resistance, characterized by increased EMT markers. This highlights the need for strategies targeting EMT pathways.

### 2.2. Chronic Trastuzumab Exposure Activates the Hedgehog Pathway

To understand the EMT-regulating mechanism associated with trastuzumab resistance, the Hedgehog pathway, a major regulator of EMT, was analyzed. The expression of Hedgehog-responsive genes (*BMP4*, *Gli1*, *Gli2*, *Hhip*, *Ptch1*, *Ptch2*) in SKBR3 and HCC1954 cells was examined over time to observe the short-term effects of trastuzumab. Both cell lines were treated daily with 0.05 μM trastuzumab, and gene expression was assessed on days 2, 4, and 15.

All markers significantly decreased on day 2 and remained unchanged on days 4 and 15 in both SKBR3 and HCC1954 cells (Figure 2A,D). However, in trastuzumab-resistant SKBR3 and HCC1954 cells, all markers significantly increased (Figure 2B,E). Additionally, acquired resistance led to significant activation of the Hedgehog pathway in both cell lines, as demonstrated by a luciferase reporter assay (Figure 2C,F). These findings suggest that trastuzumab treatment initially suppresses the expression of Hedgehog-responsive genes in HER2-positive breast cancer cells. However, long-term exposure and the development of resistance result in significant activation of the Hedgehog pathway.

### 2.3. Hedgehog Pathway Controls EMT in HER2-Positive Trastuzumab-Resistant Cells

To understand the role of the Hedgehog pathway in EMT and trastuzumab resistance, GANT61 was used as a Hedgehog pathway inhibitor, and SAG21K was used as a Hedgehog activator. The effects of GANT61 and SAG21K on Hedgehog-responsive genes were initially validated (Appendix A).

SKBR3 and HCC1954 cells were then treated with GANT61 and SAG21K for 24 h to observe how Hedgehog pathway modulation affects EMT. Treatment with GANT61 resulted in a decreased expression of EMT markers (*Slug*, *Snail*, *Twist2*, *Zeb1*) (Figure 3A,B,E,F), while treatment with SAG21K led to an increased expression of these markers (Figure 3C,D,G,H) in both parental and resistant SKBR3 and HCC1954 cells. These findings highlight the potential of targeting the Hedgehog pathway to influence EMT dynamics in HER2-positive breast cancer.

### 2.4. Integrin β3 Regulates the Hedgehog Pathway, While the Hedgehog Pathway Does Not Regulate ITGB3

In our previous study, increased ITGB3 gene expression was observed, indicating its involvement in regulating stemness through the Notch pathway in HER2-positive breast cancer cell lines. Given the well-known crosstalk between the Notch and Hedgehog pathways, the possible involvement of ITGβ3 in the Hedgehog pathway was investigated. Hedgehog-responsive gene expressions and activity were analyzed upon overexpression and silencing of ITGβ3 in both SKBR3 and HCC1954 cells (Appendix A). An increase in Hedgehog-responsive gene expressions (Figure 4A,B,G,H) and Gli reporter activity (Figure 4C,I) was observed with ITGβ3 overexpression, whereas silencing of ITGβ3 resulted in decreased gene expressions (Figure 4D,E,J,K) and Gli reporter activity (Figure 4F,L) in SKBR3 and HCC1954 parental and resistant cells.

In our previous study [16], it was observed that ITGβ3 gene expression increased and was involved in regulating stemness through the Notch pathway in HER2-positive breast cancer cell lines. Given the known interaction between the Notch and Hedgehog pathways [17], the potential role of ITGβ3 in the Hedgehog pathway was investigated.

Hedgehog-responsive gene expressions and activity were analyzed following ITGβ3 overexpression and silencing in both SKBR3 and HCC1954 cells (Appendix A). ITGβ3 overexpression led to increased Hedgehog-responsive gene expressions (Figure 4A,B,G,H) and Gli reporter activity (Figure 4C,I). Conversely, silencing ITGβ3 resulted in decreased gene expressions (Figure 4D,E,J,K) and Gli reporter activity (Figure 4F,L) in both SKBR3 and HCC1954 parental and resistant cells.

A cumulative increase in Hedgehog-responsive gene expressions was observed in both parental and resistant SKBR3 and HCC1954 cells when ITGβ3 was overexpressed in the presence of SAG21K. Conversely, a cumulative decrease was noted when ITGβ3 was silenced in the presence of GANT61 (Appendix A).

Interestingly, even with GANT61 treatment, ITGβ3 overexpression was able to induce the Hedgehog-responsive pathway in both parental and resistant SKBR3 and HCC1954 cells, compared to GANT61 treatment alone. This observation suggests that ITGβ3 may activate the Hedgehog pathway through an alternative mechanism or crosstalk, bypassing the effects of smoothened and Gli proteins.

Additionally, when ITGβ3 was silenced, Hedgehog markers decreased even in the presence of SAG21K, compared to SAG21K treatment alone (Figure 4M–P). ITGβ3 expression was analyzed in the presence of GANT61 and SAG21K, revealing that neither treatment altered ITGβ3 expression (Figure 4Q,R). These findings highlight ITGβ3 as a potential regulator of the Hedgehog pathway in HER2-positive breast cancer cells, affecting both the baseline pathway activity and response to Hedgehog modulators like SAG21K and GANT61.

### 2.5. ITGβ3 Regulates EMT

The role of ITGβ3 in EMT was investigated by analyzing EMT marker expressions following ITGβ3 overexpression and silencing. ITGβ3 overexpression led to increased expressions of *Slug*, *Snail*, *Twist2*, and *Zeb1* in both parental and resistant SKBR3 and HCC1954 cells (Figure 5A,B,E,F). Conversely, ITGβ3 silencing resulted in decreased expressions of these markers (Figure 5C,D,G,H). These findings suggest that ITGβ3 plays a significant role in promoting EMT in HER2-positive breast cancer cells, affecting both the baseline levels of EMT markers and their changes in response to resistance.

### 2.6. ITGβ3 Controls EMT Independently of the Hedgehog Pathway

To observe the interaction between Hedgehog signaling and ITGβ3 in EMT, the effects of ITGβ3 overexpression and silencing, combined with Hedgehog activators and inhibitors, were analyzed for their impact on EMT gene expression.

It was observed that EMT markers increased in both parental and resistant SKBR3 and HCC1954 cells when ITGβ3 was overexpressed in the presence of SAG21K (Appendix A). In contrast, EMT markers decreased when ITGβ3 was silenced in the presence of GANT61 (Appendix A).

When ITGβ3 was overexpressed in the presence of GANT61 or silenced in the presence of SAG21K, it was found that GANT61 alone reduced the expressions of *Slug*, *Snail*, *Twist2*, and *Zeb1*. However, despite this suppression, significant increases in these EMT markers were observed with ITGβ3 overexpression compared to the control. On the other hand, while SAG21K alone increased *Slug*, *Snail*, *Twist2*, and *Zeb1* expressions, silencing ITGβ3 led to decreased EMT marker levels even in the presence of SAG21K (Figure 6A,D).

### 2.7. Trastuzumab + Cilengitide Combination Decreases the Hedhehog Pathway

To address EMT development and drug resistance, ITGβ3, which regulates both Hedgehog and EMT gene expressions, was targeted. SKBR3 and HCC1954 cells were treated with trastuzumab (IC_50_/4), cilengitide (IC_50_/4; an integrin inhibitor targeting αvβ3 and α5β1 integrins), and a combination of trastuzumab and cilengitide. Treatments were applied daily, and the results were analyzed on days 2, 4, and 15.

Trastuzumab monotherapy decreased Hedgehog markers on day 2, but these levels remained unchanged on day 4 (Figure 7B and Figure 8B). By day 15, a significant increase in Hedgehog marker gene expressions was observed in both SKBR3 and HCC1954 parental cells. No changes in Hedgehog marker genes were noted in SKBR3- and HCC1954-resistant cells on days 2, 4, and 15 during trastuzumab monotherapy (Figure 7F and Figure 8F).

Cilengitide monotherapy showed no significant changes on day 2 in either SKBR3 or HCC1954 parental cells. However, a significant decrease in BMP4, Gli1, Gli2, Ptch1, and Ptch2 was observed on day 4, and a decrease in Hhip was noted on day 15 in SKBR3 parental cells. All EMT markers decreased in HCC1954 parental cells (Figure 7C and Figure 8C). In both SKBR3- and HCC1954-resistant cells, no changes were observed on day 2, but significant decreases were noted on days 4 and 15 (Figure 7G and Figure 8G).

In the trastuzumab + cilengitide combination therapy, all EMT markers significantly decreased on day 2 and remained low on day 4. Remarkably, these markers continued to decrease significantly on day 15 in both SKBR3 and HCC1954 parental and resistant cells (Figure 6D,H and Figure 7D,H).

### 2.8. Trastuzumab + Cilengitide Combination Decreases EMT Marker Expressions

To analyze the impact of the trastuzumab + cilengitide combination on EMT marker gene expressions, SKBR3 and HCC1954 cells were treated with IC_50_/4 trastuzumab and IC_50_/4 cilengitide as monotherapies and a combination of both drugs. Treatments were administered daily, and the results were assessed on days 2, 4, and 15.

Trastuzumab monotherapy reduced EMT marker levels on day 2, but these levels remained unchanged on day 4 (Figure 9B and Figure 10B). By day 15, however, a significant increase in EMT marker expressions was observed in both SKBR3 and HCC1954 parental cells compared to days 2 and 4. No changes in EMT markers were noted in SKBR3- and HCC1954-resistant cells on days 2, 4, and 15 during trastuzumab monotherapy (Figure 9F and Figure 10F).

Cilengitide monotherapy did not show significant changes on day 2 but resulted in a significant decrease in EMT markers on days 4 and 15 in both SKBR3 and HCC1954 parental and resistant cells (Figure 9C,G and Figure 10C,G).

In the trastuzumab + cilengitide combination therapy, EMT marker levels began to decrease on day 2 and remained low on days 4 and 15 in both SKBR3 and HCC1954 parental and resistant cells (Figure 9D,H and Figure 10D,H).

## 3. Discussion

Metastasis is the leading cause of death in HER2-positive breast cancer. It involves complex mechanisms such as EMT transition [20]. Until recently, there were no effective treatments for advanced HER2-positive breast cancer. Trastuzumab, the FDA-approved treatment for HER2-positive breast cancer, [21] initially reduces tumor size but often loses its effectiveness over the long term as tumor cells develop resistance after approximately six months [22]. Therefore, it is important to understand and develop new strategies to overcome trastuzumab resistance.

RGD-binding integrins play crucial roles in signaling and are significant contributors to tumor progression and resistance mechanisms. In healthy cells, osteopontin binding to integrin αvβ3 regulates proliferation, migration, and apoptosis. However, in breast cancer, this interaction promotes cell motility, angiogenesis, metastasis, and tamoxifen resistance [23,24]. The increased expression of the β3 integrin has been implicated in drug resistance, including resistance to erlotinib and lapatinib in lung cancer and to linsitinib in pancreatic cancer, through interactions with galectin-3 and the activation of KRAS, RELB, and the NF-ΚB pathway [8,25]. Furthermore, elevated β3 integrin levels have been shown to regulate Notch signaling, contributing to stemness in HER2-positive breast cancer [16].

Despite these insights, the role of ITGB3 in EMT and Hedgehog signaling in HER2-positive breast cancer has not been previously explored. To investigate the involvement of ITGB3 in EMT, we analyzed the expression of EMT-responsive genes in the presence and absence of ITGB3 and employed a combination therapy targeting ITGB3 to disrupt EMT. Additionally, we examined the Hedgehog pathway and assessed the impact of the combination therapy on its signaling.

In our previous study ITGB3 expression was found to significantly increase in HER2-positive breast cancer cell lines among eight RGD-binding integrins, promoting stemness through Notch signaling [16]. Given the complex interactions between the Notch and Hedgehog pathways in tumorigenesis [26], ITGβ3’s involvement in Hedgehog signaling was investigated. The overexpression of ITGβ3 led to increased Hedgehog-responsive gene expressions and Gli reporter activity, while silencing ITGβ3 resulted in decreased expressions. Notably, ITGβ3 was able to activate Hedgehog signaling even in the presence of the Hedgehog inhibitor GANT61, while its silencing reduced signaling even when stimulated with SAG21K. This finding suggests the presence of an alternative activation mechanism or additional crosstalk beyond the SMOOTHENED and GLI proteins. Modulation of the Hedgehog pathway did not affect ITGB3 expression, indicating a unidirectional regulatory relationship where ITGβ3 influences Hedgehog signaling without reciprocal regulation.

Trastuzumab initially reduces EMT, but it often loses its effectiveness over the long term as tumor cells develop resistance and begin to induce EMT [27,28]. To investigate the short- and long-term effects of trastuzumab on EMT, we compared EMT marker gene expression in HER2-positive breast cancer cells SKBR3 and HCC1954. Initially, trastuzumab treatment decreased EMT marker expressions (*Slug*, *Snail*, *Twist2*, *Zeb1*) during the early treatment phase (days 2, 4, and 15). However, after three months of chronic trastuzumab exposure, resistance developed and a significant increase was observed in EMT markers (*Slug*, *Snail*, *Twist2*, *Zeb1*). This observation suggests that trastuzumab initially suppresses EMT. However, prolonged exposure leads to adaptive changes that enhance EMT and promote resistance, consistent with the literature on EMT’s role in trastuzumab resistance. For instance, Burnett et al. demonstrated that sustained trastuzumab treatment in HER2-positive breast cancer with PTEN (phosphatase and tensin-like protein) inactivation leads to resistance, induces a mesenchymal phenotype, and increases metastatic potential [28].

Numerous studies have explored the short- and long-term effects of trastuzumab on signaling pathways. For instance, Bagnato et al. demonstrated that trastuzumab treatment for 2, 20, 45, and 120 min progressively decreased pERK1/2 expression, indicating reduced MAPK activity in SKBR3 cells [29]. Conversely, Zhuang et al. reported an increase in pERK1/2 expression in trastuzumab-resistant SKBR3 cells compared to parental cells [30]. However, the involvement of the Hedgehog pathway in this process has not been investigated before.

The Hedgehog pathway, known to regulate EMT, was analyzed in the context of HER2-positive breast cancer and trastuzumab resistance. Our results showed that trastuzumab initially suppresses Hedgehog pathway markers (*BMP4*, *Gli1*, *Gli2*, *Hhip*, *Ptch1*, *Ptch2*) in both SKBR3 and HCC1954 cells. However, in trastuzumab-resistant cells, significant activation of the Hedgehog pathway was confirmed by increased luciferase reporter activity. This novel finding suggests a mechanistic link between Hedgehog signaling and EMT in trastuzumab resistance.

Interestingly, it was found that ITGβ3 overexpression increased the expression of EMT markers (*Slug*, *Snail*, *Twist2*, *Zeb1*), while ITGβ3 silencing decreased these markers. This effect persisted even in the presence of the Hedgehog inhibitor GANT61 and activator SAG21K. This finding suggests that ITGβ3 may regulate EMT through mechanisms independent of the Hedgehog pathway. These findings emphasize the role of EMT in trastuzumab resistance and highlight the need for therapeutic strategies targeting EMT-associated pathways to improve outcomes for HER2-positive breast cancer patients. 

Targeting integrins with small molecules or antibodies has been explored as a potential strategy to overcome drug resistance in various cancers. Inhibitors of integrins αvβ3 and α5β1, such as cilengitide, have shown promising results in preclinical studies. For example, β3 integrin interaction with KRAS has been linked to a resistance to EGFR inhibition, which can be reversed by inhibiting β3 signaling [8]. Resistance to other targeted agents has been mitigated using β3 integrin antagonists or through β3 knockdown [31,32]. Moreover, combining an anti-αv integrin antibody with Src kinase inhibition has enhanced antiproliferative and antimigratory effects in colon cancer cell lines [33]. This increased antimigratory activity due to the addition of an integrin inhibitor to the combination therapy could be an effective antimetastatic strategy to overcome drug resistance.

Given ITGβ3’s promising therapeutic potential, trastuzumab and cilengitide were evaluated both as monotherapies and a combination therapy. Trastuzumab monotherapy initially led to a transient reduction in Hedgehog and EMT markers, but these markers increased with prolonged treatment. Cilengitide monotherapy resulted in a sustained decrease in both Hedgehog and EMT markers over time. Notably, the combination therapy of trastuzumab and cilengitide resulted in a significant and sustained reduction in Hedgehog and EMT markers in both parental and resistant cell lines. This finding indicates that targeting ITGβ3 in combination with trastuzumab effectively suppresses both Hedgehog signaling and EMT.

Many clinical trials have been conducted to evaluate cilengitide’s antitumour effects and dose safety. A phase II study (EMD009) was conducted with the participation of 81 patients who had recurrent glioblastoma. In patients treated with 2000 mg of cilengitide as a monotherapy, the 6-month progression-free survival was 15%, and the median overall survival was 9.9 months [34]. These hopeful results led to phase III studies, but the addition of cilengitide to standard therapies did not improve the outcomes [35].

Cilengitide has also been studied in combination therapies. In A549 and H1299 non-small lung cancer cells, erlotinib as a monotherapy effectively decreased proliferation, both canonical and non-canonical TGF-β pathways, and TGF-β induced EMT markers [36]. The addition of cilengitide to erlotinib decreased proliferation and EMT markers more than the erlotinib monotherapy. Another study showed that gefitinib efficiently decreased the proliferation and TGF-β induced EMT markers such as vimentin and e-cadherin in A549 non-small lung cancer cells, and cilengitide also as a monotherapy decreased proliferation, but the combination of gefitinib + cilengitide increased the efficacy of gefitinib and enhanced its effects on TGF-β signaling and EMT markers [37]. 

So, in this aspect, using cilengitide in combination therapies to maximize the benefits of the drugs might be beneficial to decrease the potential pro-tumorigenic effect of cilengitide [38].

Clinically, trastuzumab combined with cilengitide therapy could be particularly beneficial for patients with HER2-positive breast cancer who develop resistance to trastuzumab. By targeting the integrins that regulate EMT and the Hedgehog pathway, it may be possible to prevent or delay the progression of resistance, thereby improving patient outcomes. This approach could be integrated into current treatment regimens to enhance the efficacy of trastuzumab and provide a more durable response.

However, while ITGβ3 has been identified as a regulator of Hedgehog signaling and EMT, the exact molecular mechanisms remain unclear and warrant deeper investigation. To understand the limits of the combination therapy, it is also necessary to analyze the efficacy of the treatment based on subtypes of HER2 overexpression, such as HER2-enriched (HR−/HER2+) and triple-positive (HR+/HER2+)) [39]. Animal modeling studies and clinical trials are needed to determine optimal dosing, scheduling, and biomarkers and to assess long-term effects and potential resistance.

Despite the promising results, additional research and clinical validation are essential to confirm the therapeutic potential and safety of targeting ITGβ3 in combination with trastuzumab for HER2-positive breast cancer.

## 4. Conclusions

Our study shows that chronic trastuzumab exposure leads to EMT and Hedgehog pathway activation, contributing to resistance in HER2-positive breast cancer cells. ITGB3 is crucial in regulating both Hedgehog signaling and EMT, and its inhibition combined with trastuzumab presents a promising approach to overcoming resistance. These results highlight the need to target ITGβ3 to enhance treatment outcomes in trastuzumab-resistant breast cancer.

## 5. Methods

### 5.1. Cell Culture, Transfection, Pharmacological Treatment

HER2-positive breast cancer cell lines, HCC1954 (ATCC Cat#CRL2338) and SKBR3 (ATCC Cat#HTB30), were sourced from ATCC [40,41]. These cell lines, known for their ability to develop trastuzumab resistance, were cultured in DMEM medium supplemented with 10% FBS, 1% sodium pyruvate, and 2 mM L-glutamine. To generate trastuzumab-resistant variants, HCC1954 and SKBR3 cells were subjected to escalating doses of trastuzumab (0.1–10 μM) over a three-month period. Acquisition of resistance was validated via MTT viability assays, revealing increased IC50 values from approximately 0.2 to 2.6 μM in SKBR3 cells and from 0.3 to 2.4 μM in HCC1954 cells.

Cells were seeded into six-well plates (2.2 × 10^5^/well) at 40–60% confluency and transfected 24 h later with 2 μg of plasmid DNA per well using Lipofectamine 2000, following the manufacturer’s instructions. The IC50 values for cilengitide were determined to be 0.8 μM for SKBR3-P, 0.6 μM for SKBR3-R, 0.6 μM for HCC1954-P, and 0.7 μM for HCC1954-R, as reported in a previous study [16]. 

### 5.2. Luciferase Reporter Assay

To evaluate Hedgehog pathway activation, GLI-responsive reporter plasmids were employed. SAG21K, a Hedgehog activator targeting the smoothened protein, served as the positive control [42], while GANT61, a Gli inhibitor, acted as the negative control [43,44]. The total amount of plasmid and drug per well was standardized with an empty-FLAG vector and DMSO. The pCMV-β-Gal plasmid was generously provided by Talat Nasim from the University of Bradford, UK, and the pMuLE_ENTR_12GLI-FLuc_R4-R3 plasmid was obtained from Manfred Ogris [45] (Addgene plasmid #113712; http://n2t.net/addgene:113712, accessed on 1 July 2024).

Cells were lysed 24 h post-transfection using reporter lysis buffer (Promega, Cat. No. E4030), and luciferase activity was measured using a Fluoroskan Ascent FL luminometer (Thermo Scientific-Waltham, MA, USA) immediately after adding the luciferase assay substrate. Luciferase activity was normalized to transfection efficiency using β-galactosidase activity, quantified by absorbance at 405 nm after incubation with ONPG (4 mg/mL) + β-mercaptoethanol + Z buffer (Na_2_HPO_4_·7H_2_O (0.06 M) + NaH_2_PO_4_·H_2_O (0.04 M) + 0.5 mL 1 M KCl (0.01 M) + 1 M MgSO_4_ (0.001 M)). The reaction was stopped with 1 M Na_2_CO_3_ buffer. Reporter assay data were normalized to β-galactosidase activity to account for transfection efficiency.

### 5.3. Quantitative Real-Time PCR

RNA extraction was performed using the Qiagen RNeasy kit according to the manufacturer’s instructions. cDNA synthesis was conducted with Biorad iScript Reverse Transcription Supermix (Bio-Rad, Hercules, CA, USA) for RT-qPCR. PCR amplification utilized the Bio-Rad iTaq Universal SYBR Green One-Step Kit, with Ct values measured using the Applied Biosystems ABI 7500 Real-Time Instrument (Thermo Scientific-Waltham, MA, USA) and 7500 software v1. The PCR cycle conditions were 95 °C for 10 s, 60 °C for 1 min (repeated for 40 cycles), followed by melting curve analysis (95 °C for 15 s, 60 °C for 1 min, 95 °C for 15 s). Primers for genes (GAPDH, *Slug*, *Snail*, *Twist2*, *Zeb1*, ITGB3, *BMP4*, *Gli1*, *Gli2*, *Hhip*, *Ptch1*, *Ptch2*) were sourced from Sigma (Burlington, MA, USA) KiqStart. Each PCR was run in triplicate, and the experiments were performed three times with different samples. Data were calculated according to the following formula: ΔCt = Ct (average of target gene) − Ct (average of housekeeping gene). The 2^–∆Ct^ value was calculated, and 2^–∆∆Ct^:2^–∆Ct (sample)^/2^–∆Ct(control)^ was calculated to show fold changes. 

### 5.4. Statistical Analysis

Statistical analysis was performed using a two-tailed Student’s *t*-test and two-way ANOVA. A Tukey’s post hoc test was applied to determine *p*-values following the ANOVA test. Results were considered significant at * *p* ≤ 0.05. Error bars represent the standard deviation (±SD) of three independent experiments, each conducted in triplicate.

## Figures and Tables

**Figure 1 ijms-25-08640-f001:**
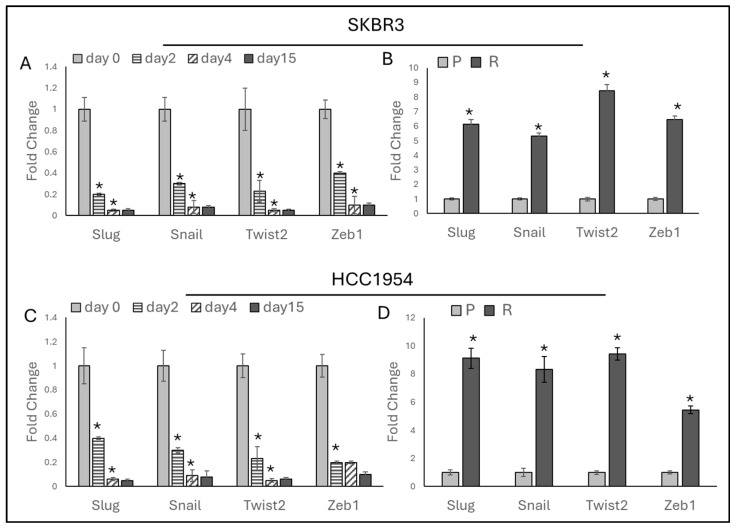
EMT markers increased by acquired trastuzumab resistance. (**A**) In SKBR3 cells, EMT-responsive markers decreased with trastuzumab treatment on days 2, 4, and 15. (**B**) In SKBR3 cells, EMT markers increased due to acquired trastuzumab resistance (90 days). (**C**) In HCC1954 cells, EMT markers decreased with trastuzumab treatment on days 2, 4, and 15. (**D**) In HCC1954 cells, EMT markers increased due to acquired trastuzumab resistance (90 days). For panels A and C, a two-way ANOVA with a Tukey’s post hoc test was used. For panels B and D, a two-tailed Student’s *t*-test was used. Parental (P) and resistant (R) cell lines are indicated. * *p* ≤ 0.05, n = 3 ± SD.

**Figure 2 ijms-25-08640-f002:**
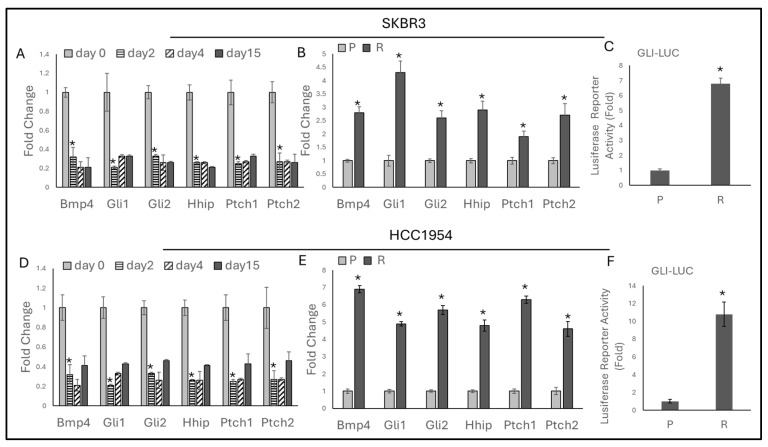
Trastuzumab treatment decreases Hedgehog pathway markers on day 2, with no changes on days 4 and 15. However, resistance activates the Hedgehog pathway. (**A**) In SKBR3 cells, Hedgehog-responsive markers decreased with trastuzumab treatment on day 2 and remained unchanged on days 4 and 15. (**B**) Trastuzumab resistance induces Hedgehog-responsive genes in SKBR3-resistant cells. (**C**) Acquired resistance activates the Hedgehog pathway in SKBR3-resistant cells. (**D**) In HCC1954 cells, Hedgehog-responsive markers decreased with trastuzumab treatment on day 2 and showed no significant change on days 4 and 15. (**E**) Trastuzumab resistance induces Hedgehog-responsive genes in HCC1954-resistant cells. (**F**) Acquired resistance activates the Hedgehog pathway in HCC1954-resistant cells. For panels A and D, a two-way ANOVA with a Tukey’s post hoc test was used. For panels B, C, E, and F, a two-tailed Student’s *t*-test was used. Parental (P) and resistant (R) cell lines are indicated. * *p* ≤ 0.05, n = 3 ± SD.

**Figure 3 ijms-25-08640-f003:**
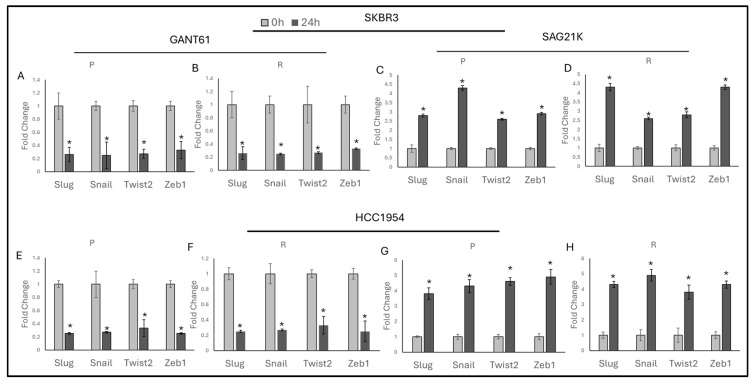
Hedgehog controls EMT in both parental and resistant cells. GANT61 decreases EMT markers in (**A**) SKBR3-P, (**B**) SKBR3-R cells and (**E**) HCC1954-P, (**F**) HCC1954-R cells. SAG21K increases EMT markers in (**C**) parental SKBR3 cells, (**D**) resistant SKBR3 cells (**G**), parental HCC1954 cells, and (**H**) resistant HCC1954 cells. P: parental, R: resistant; a two-tailed Student’s *t*-test was used. * *p* ≤ 0.05, n = 3 ± SD.

**Figure 4 ijms-25-08640-f004:**
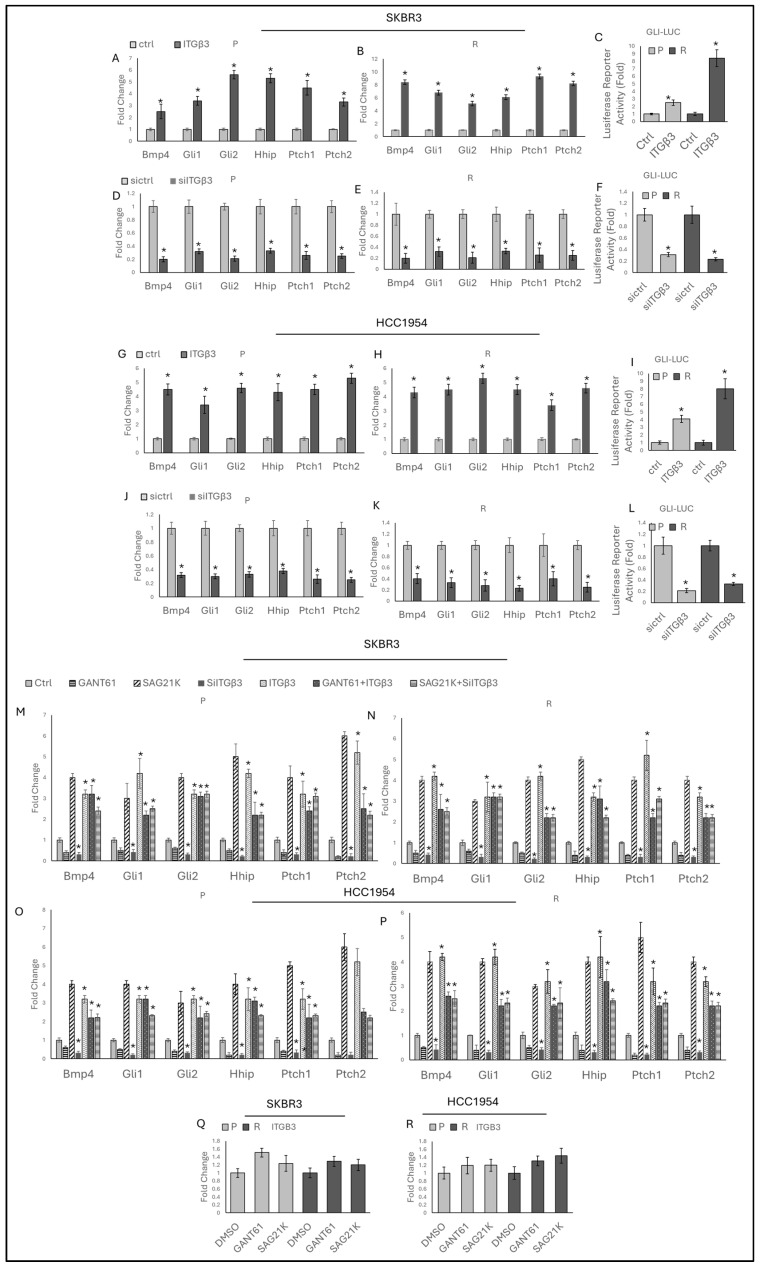
ITGβ3 controls the Hedgehog pathway, but the Hedgehog pathway does not control ITGβ3. Overexpression of ITGβ3 increases Hedgehog-responsive gene expressions in (**A**) SKBR3-P, (**B**) SKBR3-R, (**G**) HCC1954-P, and (**H**) HCC1954-R cells, Silencing of ITGβ3 decreases Hedgehog-responsive gene expressions in (**D**) SKBR3-P, (**E**) SKBR3-R, (**J**) HCC1954-P, and (**K**) HCC1954-R cells. Overexpression of ITGβ3 activates the Hedgehog pathway in parental and resistant (**C**) SKBR3 and (**I**) HCC1954 cells. Silencing of ITGβ3 suppresses the Hedgehog pathway in parental and resistant (**F**) SKBR3 and (**L**) HCC1954 cells. Hedgehog marker expressions were analyzed in the presence of GANT61, SAG21K, ITGβ3 silencing, ITGβ3 overexpression, ITGβ3 overexpression in the presence of GANT61, and ITGβ3 silencing in the presence of SAG21K in (**M**) SKBR3-P, (**N**) SKBR3-R, (**O**) HCC1954-P, and (**P**) HCC1954-R cells. *ITGB3* gene expression was analyzed in the presence of GANT61 and SAG21K in parental and resistant (**Q**) SKBR3 and (**R**) HCC1954 cells. P: parental, R: resistant. In panel A–L, a two-tailed Student’s *t*-test was used, and in panel M–P, a two-way ANOVA variation test and Tukey’s post hoc test were used. * *p* ≤ 0.05, n = 3 ± SD.

**Figure 5 ijms-25-08640-f005:**
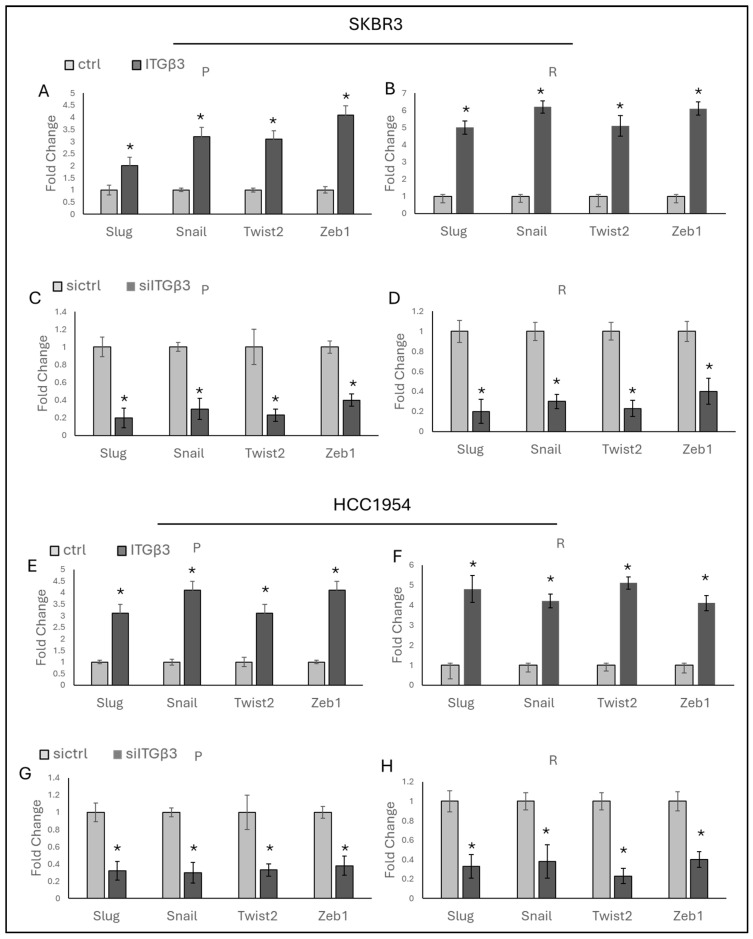
ITGβ3 regulates EMT. (**A**) Overexpression of ITGβ3 increases EMT-responsive gene expressions in (**A**) SKBR3-P and (**B**) SKBR3-R and (**E**) HCC1954-P and (**F**) HCC1954-R cells. Silencing of ITGβ3 decreases Hedgehog-responsive gene expressions in (**C**) SKBR3-P and (**D**) SKBR3-R and (**G**) HCC1954-P and (**H**) HCC1954-R cells P: parental, R: resistant; a two-tailed Student’s *t*-test was used. * *p* ≤ 0.05, n = 3 ± SD.

**Figure 6 ijms-25-08640-f006:**
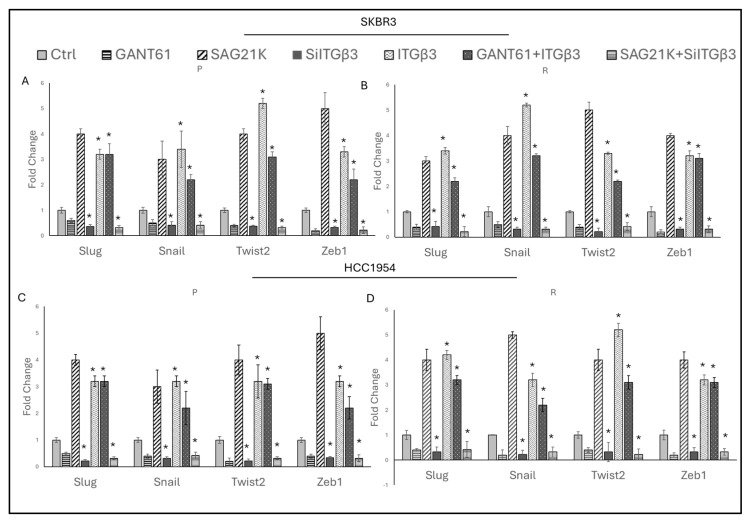
ITGβ3 controls EMT independently of the Hedgehog pathway. EMT marker expressions were analyzed in the presence of GANT61, SAG21K, ITGβ3 silencing, ITGβ3 overexpression, ITGβ3 overexpression in the presence of GANT61, and ITGβ3 silencing in the presence of SAG21K in (**A**) SKBR3-P, (**B**) SKBR3-R, (**C**) HCC1954-P, and (**D**) HCC1954-R cells. A two-way ANOVA variation test and Tukey’s post hoc test were used. * *p* ≤ 0.05, n = 3 ± SD.

**Figure 7 ijms-25-08640-f007:**
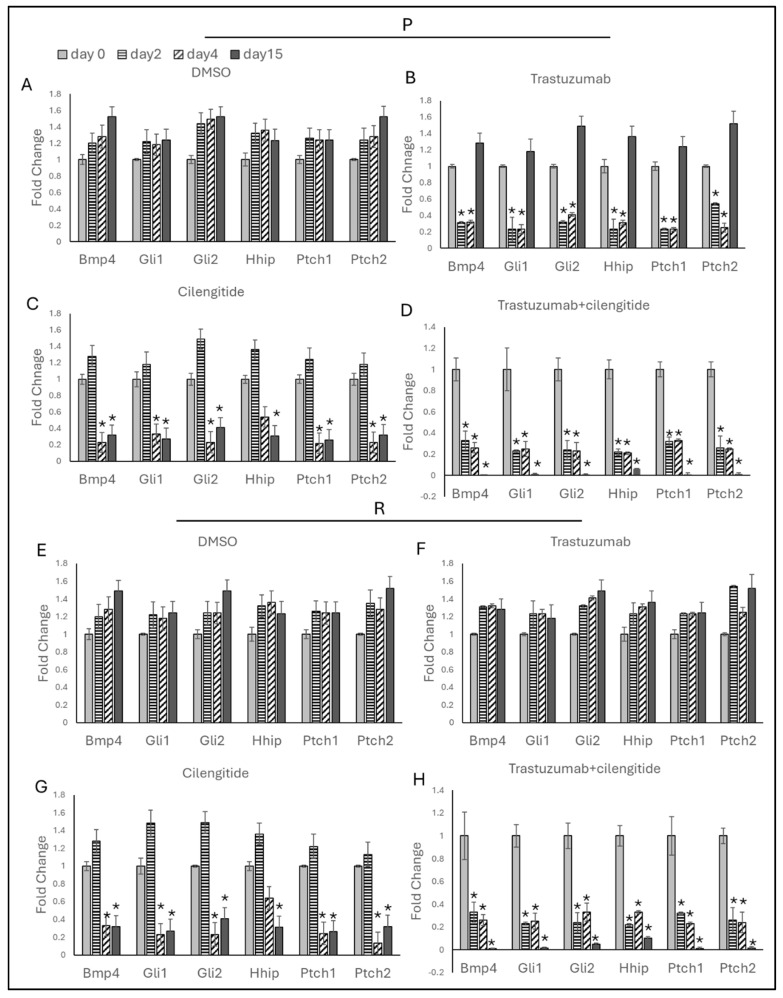
Trastuzumab + cilengitide chronic treatment decreases the Hedgehog pathway in SKBR3 parental and resistant cells. Hedgehog-responsive genes analyzed on days 2, 4, and 15 with (**A**) DMSO as the control, (**B**) trastuzumab and (**C**) cilengitide monotherapy, and (**D**) trastuzumab + cilengitide combination therapy in parental SKBR3 cells and with (**E**) DMSO as the control, (**F**) trastuzumab and (**G**) cilengitide monotherapy, and (**H**) trastuzumab+cilengitide combination therapy in resistant SKBR3 cells. A two-way ANOVA variation test and Tukey’s post hoc test were used. * *p* ≤ 0.05, n = 3 ± SD.

**Figure 8 ijms-25-08640-f008:**
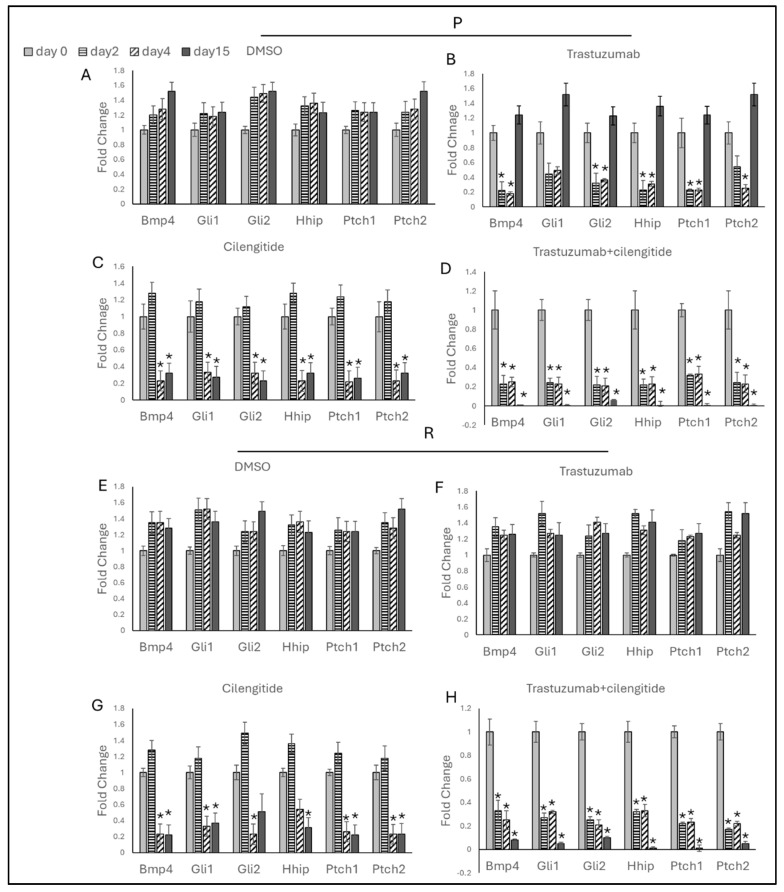
Trastuzumab + cilengitide chronic treatment decreases the Hedgehog pathway in HCC1954 parental and resistant cells. Hedgehog-responsive genes analyzed on days 2, 4, and 15 with (**A**) DMSO as the control, (**B**) trastuzumab and (**C**) cilengitide monotherapy, and (**D**) trastuzumab + cilengitide combination therapy in parental HCC1954 cells and with (**E**) DMSO as the control, (**F**) trastuzumab and (**G**) cilengitide monotherapy, and (**H**) trastuzumab + cilengitide combination therapy in resistant HCC1954 cells. A two-way ANOVA variation test and Tukey’s post hoc test were used. * *p* ≤ 0.05, n = 3 ± SD.

**Figure 9 ijms-25-08640-f009:**
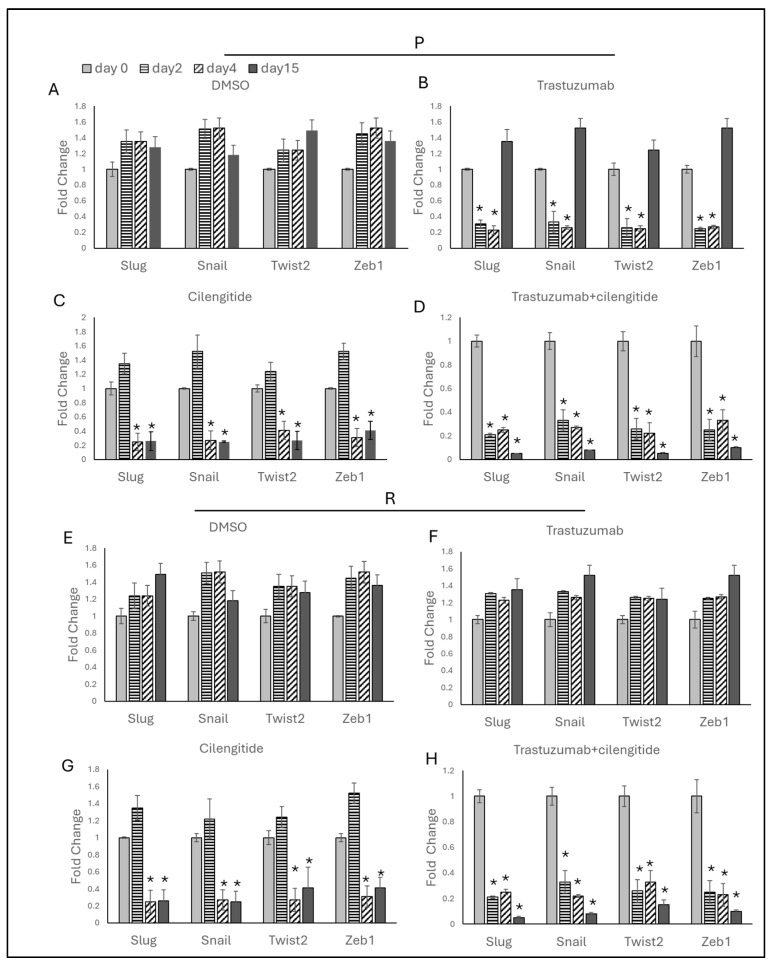
Trastuzumab + cilengitide chronic treatment decreases EMT in the parental and resistant cells of SKBR3. EMT-responsive genes analyzed on days 2, 4, and 15 with (**A**) DMSO as the control, (**B**) trastuzumab and (**C**) cilengitide monotherapy, and (**D**) trastuzumab+cilengitide combination therapy in parental SKBR3 cells and with (**E**) DMSO as the control, (**F**) trastuzumab and (**G**) cilengitide monotherapy, and (**H**) trastuzumab + cilengitide combination therapy in resistant SKBR3 cells. A two-way ANOVA variation test and Tukey’s post hoc test were used. * *p* ≤ 0.05, n = 3 ± SD.

**Figure 10 ijms-25-08640-f010:**
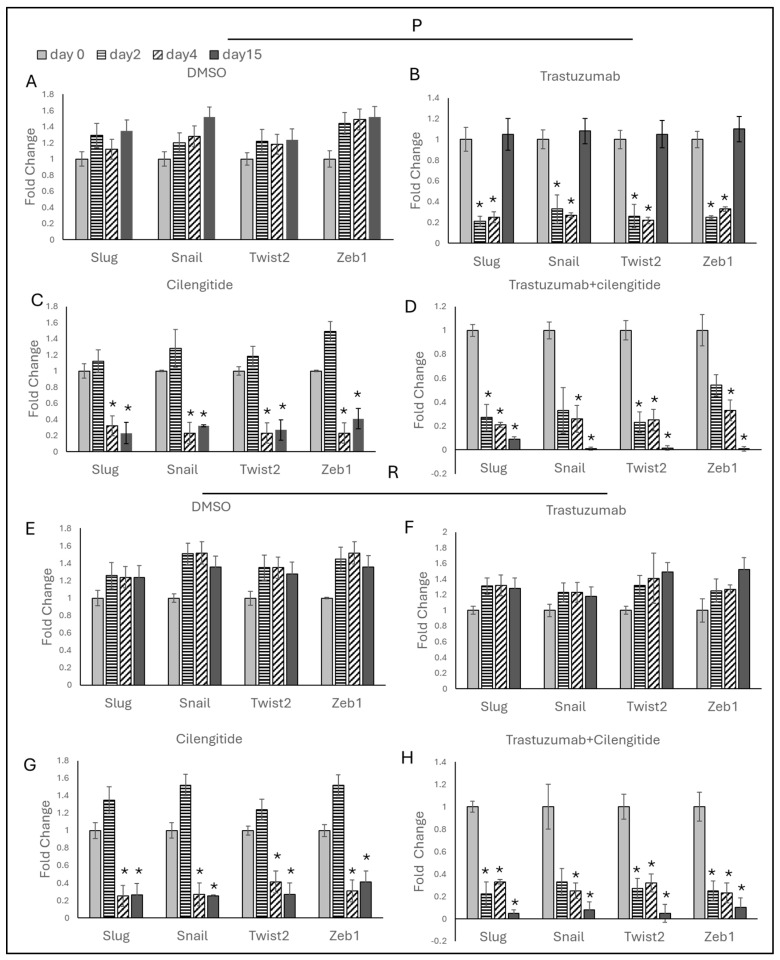
Trastuzumab + cilengitide chronic treatment decreases EMT in the parental and resistant cells of HCC1954. Hedgehog-responsive genes analyzed on days 2, 4, and 15 with (**A**) DMSO as the control, (**B**) trastuzumab and (**C**) cilengitide monotherapy, and (**D**) trastuzumab+cilengitide combination therapy in parental HCC1954 cells and with (**E**) DMSO as the control, (**F**) trastuzumab and (**G**) cilengitide monotherapy, and (**H**) trastuzumab + cilengitide combination therapy in resistant HCC1954 cells. A two-way ANOVA variation test and Tukey’s post hoc test were used. * *p* ≤ 0.05, n = 3 ± SD.

## Data Availability

The datasets and materials used and/or analyzed during the current study are available from the corresponding author upon reasonable request.

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
