# Peer review of "Targeting ITGβ3 to Overcome Trastuzumab Resistance through Epithelial–Mesenchymal Transition Regulation in HER2-Positive Breast Cancer"

_ijms, 2024, doi:10.3390/ijms25168640_

Round 1

Reviewer 1 Report

Comments and Suggestions for Authors

The article "Targeting ITGβ3 to Overcome Trastuzumab Resistance Through EMT Regulation in HER2-Positive Breast Cancer" by Asiye Boz Er and Idris Er provides valuable insights. However, there are several areas that require attention:

1. The authors should rewrite the abstract to clearly state the background, aim of the study, materials and methods, results, and conclusion without using excessive sentences.

2. The manuscript contains significant grammatical and typographical errors. Authors from non-English speaking countries should ensure that their articles are corrected by a native English speaker for grammatical, stylistic, and typographical errors.

3. The authors could include a comprehensive discussion based on the implications of the study's findings and potential clinical applications.

4. The authors could include potential side effects or risks associated with the proposed combination therapy.

5. The authors could include potential limitations of the study, and future research directions could be mentioned.

6. The study did not perform any in vivo experiments, and the HER2 xenograft animal models did not yield consistent outcomes.

7. While combination therapies are beneficial in overcoming drug resistance, the authors could also discuss potential drawbacks or side effects of combining multiple drugs.

8. Comprehensive studies are required to authenticate potent therapeutic agents based on in vivo and clinical studies, including toxicity studies, biocompatibility, biodegradability, stability, half-life in the portal circulation, renal clearance, accumulation, and uptake of drugs in targeted cancer tissues, among others.

Comments on the Quality of English Language

Extensive editing of English language required

Author Response

First of all, we wanted to express our sincere gratitude for taking the time to review our manuscript. Your detailed and insightful feedback is greatly appreciated. Your comments were incredibly helpful in highlighting areas that need improvement.

Comments and Suggestions for Authors

  1. The authors should rewrite the abstract to clearly state the background, aim of the study, materials and methods, results, and conclusion without using excessive sentences.

-The last version of abstract prepared as.

HER2-positive breast cancer, which represents 15-20% of all breast cancer cases, often develops resistance to the HER2-targeted therapy trastuzumab. Unfortunately, effective treatments for advanced HER2-positive breast cancer remain scarce. This study aims to investigate the roles of ITGβ3, and Hedgehog signalling in trastuzumab resistance and explore the potential of combining trastuzumab with cilengitide as a therapeutic strategy. Quantitative gene expression analysis was performed to assess the transcription of EMT (epithelial-mesenchymal transition) markers Slug, Snail, Twist2, and Zeb1 in trastuzumab-resistant HER2-positive breast cancer cells. The effects of ITGβ3 and Hedgehog signalling were investigated. Additionally, the combination therapy of trastuzumab and cilengitide was evaluated. Acquired trastuzumab resistance induced the transcription of Slug, Snail, Twist2, and Zeb1, indicating increased EMT. This increased EMT was mediated by ITGB3 and Hedgehog signalling. ITGβ3 regulated both the Hedgehog pathway and EMT, with the latter being independent of the Hedgehog pathway. The combination of trastuzumab and cilengitide showed a synergistic effect, reducing both EMT and Hedgehog pathway activity. Targeting ITGβ3 with cilengitide, combined with trastuzumab, effectively suppresses the Hedgehog pathway and EMT, offering a potential strategy to overcome trastuzumab resistance and improve outcomes for HER2-positive breast cancer patients.

  1. The manuscript contains significant grammatical and typographical errors. Authors from non-English speaking countries should ensure that their articles are corrected by a native English speaker for grammatical, stylistic, and typographical errors.

-All corrected by native English speaker from University of Bradford.

  1. The authors could include a comprehensive discussion based on the implications of the study's findings and potential clinical applications.

-Information which include clinical clinical trials related with our study added to improve our discussion such as;

Many clinical trials have been done to evaluate cilengitide’s antitumour effects and dose safety. A phase II study (EMD009) was done with the participation of 81 patients who has recurrent glioblastoma. In patients treated with 2000 mg of cilengitide as monotherapy, the 6 months progression free survival was 15% and median overall survival was 9.9 months (Reardon, Fink et al. 2008). These hopeful results led to phase III studies but addition of cilengitide to standard therapies did not improve the outcomes (Stupp, Hegi et al. 2014).

Cilengitide has also been studied in combination therapies. In A549 and H1299 non-small-lung cancer cells, erlotinib as a monotherapy effectively decreased proliferation, both canonical and non-canonical TGF-β pathway and TGF-β induced EMT markers (Jeong and Kim 2022). Addition of cilengitide to erlotinib decreased proliferation and EMT markers more than erlotinib monotherapy. Another study showed that gefitinib efficiently decrease the proliferation and TGF-β induced EMT markers such as vimentin and e-cadherin in A549 non-small-lung cancer cells and cilengitide also as a monotherapy decreased proliferation but combination of gefitinib+cilengitide increased the efficacy of gefitinib and enhanced its effects on TGF-β signalling and EMT markers (Jeong and Kim 2021).

 So in this aspect using cilengitide in combination therapies to maximize the benefits of the drugs or combining with a VEGF targeting inhibitor might be beneficial to decrease the potential pro-tumourigenic effect of cilengitide (Reardon and Cheresh 2011).

Clinically, trastuzumab+cilengitide combination therapy could be particularly beneficial for patients with HER2-positive breast cancer who develop resistance to trastuzumab. By targeting the integrins that regulate EMT and the Hedgehog pathway, it may be possible to prevent or delay the progression of resistance, thereby improving patient outcomes. This approach could be integrated into current treatment regimens to enhance the efficacy of trastuzumab and provide a more durable response.

  1. The authors could include potential side effects or risks associated with the proposed combination therapy.

-Addressing potential side effects or risks associated with the proposed combination therapy is challenging based on our current findings, as our study primarily focuses on the RNA responses of HER2-positive breast cancer cells. However, it is important to note that cilengitide has been generally well-tolerated in clinical settings. Such as; in a phase II clinical trial NCT00082875, cilengitide was tested on 56 patients who has unresectable stage III-IV metastatic melanoma (NIH-US, 2012). It observed that cilengitide was well tolerated by patients but had minimal clinical efficacy as a monotherapy for metastatic melanoma (Kim et al., 2012).

  1. The authors could include potential limitations of the study, and future research directions could be mentioned.

-We added the information below to clarify the potential limitations of the study and suggest directions for future research.

However, while ITGB3 has been identified as a regulator of Hedgehog signaling and EMT, the exact molecular mechanisms remain unclear and warrant deeper investigation. Animal modeling studies and clinical trials are needed to determine optimal dosing, scheduling, biomarkers, and to assess long-term effects and potential resistance. Despite the promising results, additional research and clinical validation are essential to confirm the therapeutic potential and safety of targeting ITGβ3 in combination with trastuzumab for HER2-positive breast cancer.

  1. The study did not perform any in vivo experiments, and the HER2 xenograft animal models did not yield consistent outcomes.

- Our future project aims to generate HER2 xenograft animal models to understand the dose effects of the combination therapy on HER2-positive tumors. This manuscript provides data that serves as both proof and guidance for in vivo experiments to analyze the combination and its long-term effects on mouse models. Due to budget limitations, we were unable to perform additional experiments. However, we hope that the publication of our results will help us secure more grants for further in vivo studies.

  1. While combination therapies are beneficial in overcoming drug resistance, the authors could also discuss potential drawbacks or side effects of combining multiple drugs.

- A paragraph was added to the discussion section about the addition of cilengitide to current therapies as a combination treatment and its potential results.

  1. Comprehensive studies are required to authenticate potent therapeutic agents based on in vivo and clinical studies, including toxicity studies, biocompatibility, biodegradability, stability, half-life in the portal circulation, renal clearance, accumulation, and uptake of drugs in targeted cancer tissues, among others.

- We fully agree with the reporter's insights. Looking ahead, we aim to expand our research efforts and are open to potential collaborations.

Comments on the Quality of English Language

Extensive editing of English language required

-Done.

Jeong, J. and J. Kim (2021). "Cyclic RGD Pentapeptide Cilengitide Enhances Efficacy of Gefitinib on TGF-beta1-Induced Epithelial-to-Mesenchymal Transition and Invasion in Human Non-Small Cell Lung Cancer Cells." Front Pharmacol 12: 639095.

Jeong, J. and J. Kim (2022). "Combination Effect of Cilengitide with Erlotinib on TGF-beta1-Induced Epithelial-to-Mesenchymal Transition in Human Non-Small Cell Lung Cancer Cells." Int J Mol Sci 23(7): 3423.

Reardon, D. A. and D. Cheresh (2011). "Cilengitide: a prototypic integrin inhibitor for the treatment of glioblastoma and other malignancies." Genes Cancer 2(12): 1159-1165.

Reardon, D. A., K. L. Fink, T. Mikkelsen, T. F. Cloughesy, A. O'Neill, S. Plotkin, M. Glantz, P. Ravin, J. J. Raizer, K. M. Rich, D. Schiff, W. R. Shapiro, S. Burdette-Radoux, E. J. Dropcho, S. M. Wittemer, J. Nippgen, M. Picard and L. B. Nabors (2008). "Randomized phase II study of cilengitide, an integrin-targeting arginine-glycine-aspartic acid peptide, in recurrent glioblastoma multiforme." J Clin Oncol 26(34): 5610-5617.

Stupp, R., M. E. Hegi, T. Gorlia, S. C. Erridge, J. Perry, Y. K. Hong, K. D. Aldape, B. Lhermitte, T. Pietsch, D. Grujicic, J. P. Steinbach, W. Wick, R. Tarnawski, D. H. Nam, P. Hau, A. Weyerbrock, M. J. Taphoorn, C. C. Shen, N. Rao, L. Thurzo, U. Herrlinger, T. Gupta, R. D. Kortmann, K. Adamska, C. McBain, A. A. Brandes, J. C. Tonn, O. Schnell, T. Wiegel, C. Y. Kim, L. B. Nabors, D. A. Reardon, M. J. van den Bent, C. Hicking, A. Markivskyy, M. Picard, M. Weller, R. European Organisation for, C. Treatment of, C. Canadian Brain Tumor and C. s. team (2014). "Cilengitide combined with standard treatment for patients with newly diagnosed glioblastoma with methylated MGMT promoter (CENTRIC EORTC 26071-22072 study): a multicentre, randomised, open-label, phase 3 trial." Lancet Oncol 15(10): 1100-1108.

Reviewer 2 Report

Comments and Suggestions for Authors

Authors present a work addressing: ‘Targeting ITGβ3 to overcome Trastuzumab resistance Through EMT regulation in HER2-positive breast cancer’. The aim of study was to investigate the roles of ITGB3 and Hedgehog signalling in trastuzumab resistance in breast cancer were investigated, along with the potential of combining trastuzumab with cilengitide as a therapeutic strategy. The general conclusion demonstrates that targeting ITGB3 with the integrin inhibitor cilengitide, in combination with trastuzumab, effectively suppresses both the Hedgehog pathway and EMT, offering a potential therapeutic strategy to overcome resistance and improve outcomes for HER2-positive breast cancer patients. Thus I recommend publication after some major issues have been addressed:

General: well-designed study

Major points:

1. Authors should avoid abbreviations in the title of the paper.
2. I suggest to add the paragraph in the introduction related to the novelty of this study and how your outcomes can help further studies in this area.
3. Also, in discussion section please add paragraph related to clinical and practical aspects of the study. How we can applicate your results into practice?, why your work is valuable in the field?
4. Authors should provide the study's limitations and add future perspectives in the study.
5. Throughout the work, please explain abbreviations when they are first appeared.

Minor:

1. In line 205 the word study seems to be missing.
2. Minor punctuation mistakes were detected.
3. The figures 1-4 are a bit illegible.
4. The font size is too small (all figures).

Comments on the Quality of English Language

Minor editing of English language required.

Author Response

First of all, we wanted to express our sincere gratitude for taking the time to review our manuscript. Your detailed and insightful feedback is greatly appreciated. Your comments were incredibly helpful in highlighting areas that need improvement.

Major points:

1. Authors should avoid abbreviations in the title of the paper.

-Done.

  1. I suggest to add the paragraph in the introduction related to the novelty of this study and how your outcomes can help further studies in this area.

- The last paragraph of the introduction has been rewritten to incorporate the reporter's suggestions

  1. Also, in discussion section please add paragraph related to clinical and practical aspects of the study. How we can applicate your results into practice?, why your work is valuable in the field?

- In previous clinical trials, cilengitide has demonstrated promising results, particularly when used in combination therapies. Its positive impact on EMT (targeting EMT means targeting metastasis) underscores the significance of our study. To enhance clarity, we have included additional information in the manuscript.

  1. Authors should provide the study's limitations and add future perspectives in the study.

-Added, please see the last version of manuscript.

  1. Throughout the work, please explain abbreviations when they are first appeared.
    -Corrected.

Minor:

1. In line 205 the word study seems to be missing.

-Corrected and sentence refined.

  1. Minor punctuation mistakes were detected.

-Corrected

  1. The figures 1-4 are a bit illegible.4. The font size is too small (all figures).

-The font size increased in figures.

Reviewer 3 Report

Comments and Suggestions for Authors

The study entitled "Targeting ITGβ3 to Overcome Trastuzumab Resistance Through EMT regulation in HER2-Positive Breast Cancer" evaluates the role of integrin β3 (ITGβ3) and the Hedgehog pathway in trastuzumab resistance in HER2-positive breast cancer.

Chronic exposure to trastuzumab leads to the development of drug resistance, characterized by increased expression of EMT markers and activation of the Hedgehog pathway.

The study demonstrates that ITGβ3 regulates both Hedgehog signaling and EMT, independent of the pathway's traditional mediators. Combination therapy with trastuzumab and cilengitide, an integrin inhibitor, significantly reduced EMT and Hedgehog markers, suggesting a promising approach to overcoming resistance.

I have some comments:

- At the end of the Introduction section, the aims of the study are very confused. Please try to be more specific and concise and re-write;

- It would be very interesting to apply your results also in the field of de-novo metastatic breast cancer, taking into consideration a large part of HER2-positive tumors in this setting. Cite PMID: 36551722 to improve the quality of your Discussion;

- Lines 447-453 are redundant, remove them and re-write.

Author Response

First of all, we wanted to express our sincere gratitude for taking the time to review our manuscript. Your detailed and insightful feedback is greatly appreciated. Your comments were incredibly helpful in highlighting areas that need improvement.

Comments and Suggestions for Authors

- At the end of the Introduction section, the aims of the study are very confused. Please try to be more specific and concise and re-write;

Written as;

In summary, the aim of this study is to investigate the role of ITGβ3 and Hedgehog signaling in the development of trastuzumab resistance in HER2-positive breast cancer and to explore the potential of combination therapy using trastuzumab and cilengitide to overcome this resistance. The study focuses on understanding how ITGβ3 regulates EMT and its impact on Hedgehog signaling and examines whether targeting ITGβ3 with cilengitide can suppress EMT and Hedgehog pathway activity, thereby potentially improve treatment response and overcome resistance for patients with trastuzumab-resistant HER2-positive breast cancer.

- It would be very interesting to apply your results also in the field of de-novo metastatic breast cancer, taking into consideration a large part of HER2-positive tumors in this setting. Cite PMID: 36551722 to improve the quality of your Discussion;

-We greatly appreciate your suggestion, and the reference you provided is indeed very interesting. We are keen to investigate the molecular mechanisms of De Novo Metastatic Breast Cancer in the future. However, at this time, we have not yet identified a direct connection with our current research. Thank you for your valuable input.

- Lines 447-453 are redundant, remove them and re-write.

-This part is removed and more information added about limits of the study and previous clinical trials.

Round 2

Reviewer 1 Report

Comments and Suggestions for Authors

Accept in present form

Comments on the Quality of English Language

Minor editing of English language required

Author Response

Accept in present form.

Thank you very much, I really appreciate.

Reviewer 2 Report

Comments and Suggestions for Authors

The authors incorporate my suggestions, and I therefore suggest to publish the manuscript in its current form.

Author Response

The authors incorporate my suggestions, and I therefore suggest to publish the manuscript in its current form.

Thank you very much, I truly appreciate.

Reviewer 3 Report

Comments and Suggestions for Authors

The authors made some efforts into this new version of their manuscript; however, they did not take into consideration all of my comments to improve the quality of their study.

I suggest the authors to revise once again their manuscript and add all the requested comments; moreover including a limitations paragraph before their conclusions (currently missing).

Author Response

The authors made some efforts into this new version of their manuscript; however, they did not take into consideration all of my comments to improve the quality of their study.

I suggest the authors to revise once again their manuscript and add all the requested comments; moreover including a limitations paragraph before their conclusions (currently missing).

-Thank you very much for your suggestions.

Limitations of our study added to our manuscript as;

However, while ITGβ3 has been identified as a regulator of Hedgehog signaling and EMT, the exact molecular mechanisms remain unclear and warrant deeper investigation. To understand the limits of the combination therapy, it is also necessary to analyze the efficacy of treatment based on subtypes of HER2 overexpression, such as HER2-enriched (HR−/HER2+) and triple-positive (HR+/HER2+))[1]. Animal modeling studies and clinical trials are needed to determine optimal dosing, scheduling, biomarkers, and to assess long-term effects and potential resistance.

Despite the promising results, additional research and clinical validation are essential to confirm the therapeutic potential and safety of targeting ITGβ3 in combination with trastuzumab for HER2-positive breast cancer.

Also a reference; Cite PMID: 36551722 discussed and added to our manuscript to improve the quality of our work.

  1. Tinterri, C.; Sagona, A.; Barbieri, E.; Di Maria Grimaldi, S.; Jacobs, F.; Zambelli, A.; Trimboli, R.M.; Bernardi, D.; Vinci, V.; Gentile, D. Loco-Regional Treatment of the Primary Tumor in De Novo Metastatic Breast Cancer Patients Undergoing Front-Line Chemotherapy. Cancers (Basel) 2022, 14, doi:10.3390/cancers14246237.

Round 3

Reviewer 3 Report

Comments and Suggestions for Authors

The manuscript can be accepted in the present form